# Tailoring Vibrational Signature and Functionality of 2D-Ordered Linear-Chain Carbon-Based Nanocarriers for Predictive Performance Enhancement of High-End Energetic Materials

**DOI:** 10.3390/nano12071041

**Published:** 2022-03-22

**Authors:** Alexander Lukin, Oğuz Gülseren

**Affiliations:** 1Western-Caucasus Research Center, 352808 Tuapse, Russia; 2Department of Physics, Bilkent University, Ankara 06800, Turkey; gulseren@fen.bilkent.edu.tr

**Keywords:** transformative energetics, ion-assisted pulse-plasma assembling, heteroatom doping, vibrational signature, unified templates, surface acoustic waves, plasma-acoustic coupling mechanism, directed self-assembly, data-driven carbon nanomaterials genome approach

## Abstract

A recently proposed, game-changing transformative energetics concept based on predictive synthesis and preprocessing at the nanoscale is considered as a pathway towards the development of the next generation of high-end nanoenergetic materials for future multimode solid propulsion systems and deep-space-capable small satellites. As a new door for the further performance enhancement of transformative energetic materials, we propose the predictive ion-assisted pulse-plasma-driven assembling of the various carbon-based allotropes, used as catalytic nanoadditives, by the 2D-ordered linear-chained carbon-based multicavity nanomatrices serving as functionalizing nanocarriers of multiple heteroatom clusters. The vacant functional nanocavities of the nanomatrices available for heteroatom doping, including various catalytic nanoagents, promote heat transfer enhancement within the reaction zones. We propose the innovative concept of fine-tuning the vibrational signatures, functionalities and nanoarchitectures of the mentioned nanocarriers by using the surface acoustic waves-assisted micro/nanomanipulation by the pulse-plasma growth zone combined with the data-driven carbon nanomaterials genome approach, which is a deep materials informatics-based toolkit belonging to the fourth scientific paradigm. For the predictive manipulation by the micro- and mesoscale, and the spatial distribution of the induction and energy release domains in the reaction zones, we propose the activation of the functionalizing nanocarriers, assembled by the heteroatom clusters, through the earlier proposed plasma-acoustic coupling-based technique, as well as by the Teslaphoresis force field, thus inducing the directed self-assembly of the mentioned nanocarbon-based additives and nanocarriers.

## 1. Introduction

The predictive control and enhancement of the energetic materials (EM) reactivity and energy output coupled with the mutual stabilization of high-energy components are extremely important for the development of low-cost and efficient small-scale solid propulsion systems, solid microthrusters as well as the high-efficiency micropower systems.

The objective of this study is to uncover new potential opportunities for the predictive extracting of extra energy from the EM systems at the nanoscale by manipulating the combinations of the unique structural and physicochemical properties of functionalized low-dimensional nanocarbon allotropes by the deep materials informatics-based toolkit [1,2,3,4].

Over the past few decades, carbon-based nanomaterials, and especially low-dimensional nanocarbon allotropes, have revolutionized the field of EM science and technology [5,6,7,8,9,10,11]. The physicochemical properties of the carbon-based nanomaterials essentially depend on their spatial structure. Carbon atoms, due to various forms of structural hybridization, are capable of giving rise to graphite, diamond, graphene, carbyne chains and many other specific allotropes. Every specific carbon allotrope has significantly different structural, mechanical and electronic properties.

Low-dimensional nanocarbon allotropes represent promising building blocks both for nanohybrid systems and for the macroscopic assembly of emerging multifunctional composites, as they possess a unique nanoarchitecture, a set of physicochemical properties and abundant functionalities that are of great interest for high-end applications in the emerging fields of nanoscience and nanotechnology. The “holy grail” of one-dimensional carbon allotropes, carbyne, represents a one-dimensional chain of carbon atoms.

Numerous recent experimental studies prove that nanoadditives based on carbon allotropes significantly affect the thermal decomposition, ignition, combustion and reactivity of the EMs and solid propellants, and can essentially improve their combustion characteristics, thermal stability and environmental safety [5,7]. For instance, using the detonation nanodiamonds in the solid propellant composition changes the process of thermal decomposition and leads to an essential increase in the burning rate [12,13].

Designing the carbon-based micro- and nanoadditives for use in the composition of the EM and solid propellants is aimed at increasing the intensity of heat transfer, increasing the burning rate, reducing sensitivity to external influences and improving mechanical properties. Since the mechanical properties of the EM and solid propellants are directly related to ensuring their safe use, there is a need to further improve them, for example, through the predictive programming of the microstructure of the carbon-based nanoadditives.

Among the promising carbon-based additives that have been investigated for the EM performance enhancement are graphene-based fibers (GFs) [9,10,11,14] and modified thermally expandable graphite-based fibers [15,16].

Graphene is a “miracle material” with wide capabilities that have revolutionized our world [9,10,11,14]. More recently, it has been experimentally demonstrated that graphene-based nanomaterials are promising additives for improving the performance of the EMs and solid propellants. This circumstance is associated with the unique set of structural and physicochemical properties of graphene, such as low density and, at the same time, the ultra-high mechanical strength, large specific surface area, high thermal conductivity, outstanding electrical characteristics and the high level of chemical functionality. Despite the fact that nanographene has a high flexibility, at the same time it is stronger than steel.

Taking into account its large specific surface area, along with its necessary adsorption capacity, graphene can be considered as an ideal material for supporting catalysts to prevent the aggregate and to enhance the catalytic activity of the catalyst nanoadditives of the EMs and propellants. From the simple two-dimensional graphene to the complex three-dimensional graphene interconnected structure, the basic unit is the same, but they can exhibit different morphologies, micro/nanostructures and physicochemical properties through micro- and nanostructural designs and play different roles in EMs and propellants.

Relatively recent experimental studies have confirmed that the use of graphene-based nanoadditives in the composition of the EMs and solid propellants opens up the possibility of increasing the burning rate by approximately 8–10 times, which fundamentally changes the possibilities for the development of a new generation of propulsion systems for promising aerospace systems [9,10,11]. By adding the GFs into the EM matrix, the fiber arrays orientation effect can be used both for the manipulation and enhancement of the thermal conductivity within the reaction zones and for significantly improving the mechanical properties.

Due to the fact that thermally expandable graphite-based fibers have a high level of electrical conductivity, it is possible to control their spatial orientation in the required direction using an external electromagnetic field. As a result, it is possible to create micro/nanostructures with electrical or thermal conductivity in a given direction, which can be used to program and intensify the combustion process.

Application of the ion-assisted pulse-plasma functionalization and predictive heteroatom doping techniques open up a set of novel opportunities, transforming nanocarbon catalytic additives into nanohybrid systems with multifunctional properties. The key difference between nanomaterials and nanohybrids is that the nanomaterials exhibit a certain property in one material, while the nanohybrids exhibit multiple structural and physicochemical properties within one nanomaterial.

Novel functionalities of the catalytic additives based on GFs and modified thermally expandable graphite-based fibers can be unlocked when they are used within the frame-work of the transformative energetics concept, based on the EM predictive synthesis and preprocessing at the nanoscale.

## 2. Game-Changing Transformative Energetics Concept

Recently proposed game-changing transformative energetics concepts [17,18], based on the EM predictive synthesis and preprocessing at the nanoscale, are considered as a pathway towards the development of the next generation of advanced propulsion materials for future multimode solid propulsion systems and deep-space-capable small satellites.

This technological concept brings extra energy into the EM system, which goes beyond what was historically possible using conventional processing technologies and opens novel opportunities for the mutual stabilization of high-energy components inserted into the EM system. This technique allows the extraction of extra energy from the EM system at the nanoscale.

The technological chain of the transformative energetic materials (TEM) preprocessing and synthesis includes the following main stages: preprocessing the EM components with the material properties, changing at the conversion into the nanoscale; the resonant acoustic mixing (RAM) of the EM composition, leading to changes in the material properties; the final 3D printing (additive manufacturing) of the blended EM composition into the high-end EM elements and solid propellant charges.

Numerous experimental studies have confirmed that nanosized EMs can be considered as a source of extremely high heat release rates along with outstanding possibilities of burning rates for programming, reliability, extremely high burning efficiency, safety of use and reduced sensitivity to external influences [19,20]. The fundamental distinguishing feature of the nanosized EMs is a significant increase in the specific surface area and a decrease in the distances between nanocomponents, which provides a significant increase in the chemical reaction rates while simultaneously providing a reduction of the ignition delay, as well as providing the required safety level [20].

The RAM is a relatively new and highly effective mixing technique that applies programmed low-frequency and high-intensity acoustic energy for blending highly viscous materials [21,22]. The RAM is a contactless mixing technique which ensures increased process safety that affords the potential to incorporate into the EM composition a higher proportion of high-energy-density solids, including hard-to-cut materials such as nanoenergetic systems.

Three-dimensional printing allows the growing of complex and three-dimensional structures with programmed structural and physicochemical properties that give superior control over the micro- and mesoscale spatial distribution of the induction and energy release domains, as well as the nature of the energy release [23,24].

One of the important benefits of the TEM concept is connected with the possibility of additive manufacturing in the new generation of electrically activated solid propellants, or, more shortly, ePropellants, developed by Digital Solid State Propulsion, LLC, Reno, NV, USA [25,26]. In this new technique, the electric fields are used to control the ignition and extinguishment properties, and thus the throttle, of solid propellants. These EM and solid propellants, based on ionic salts, are mainly inert until an electric current is passed through them, which activates the electrochemical reactions. Currently, the ePropellant compositions include ammonium nitrate (AN) or hydroxylamine nitrate (HAN) as oxidizing components, and, in this regard, demonstrate the low-energy performance. The new approach for the development of green ePropellants with enhanced energy performance includes using ammonium dinitramide (ADN) as the main oxidizing agent.

As a fundamentally new opportunity to further improve the energy efficiency of the TEMs, we propose to use the predictive ion-stimulated pulse-plasma modification of various carbon-based allotropes, used as catalytic nanoadditives, at the stage of the preliminary preparation of components (conversion to the nanoscale) by the plasma-driven attachment of multicavity nanomatrices of the 2D-ordered linear-chain carbon, which serve as functionalizing nanocarriers of multiple heteroatom clusters.

The vacant functional nanocavities of the nanomatrices available for heteroatom doping, including various catalytic nanoagents, create new pathways of heat transfer enhancement within the EM reaction zones. In this case, as a carbon-based allotrope applied as a catalytic nanoadditive, we propose to use GFs or modified thermally expandable graphite-based fibers.

The structural features and physicochemical properties of the 2D-ordered linear-chain carbon multicavity nanomatrices, the ion-assisted pulse-plasma growth and the heteroatom doping toolkit, as well as the predictive activation and multifunctionality property tuning, will be discussed in detail in the following sections.

## 3. Features and Synthesis of the Functionalizing Nanocarriers

### 3.1. 2D-Ordered Linear-Chain Carbon as a Functionalizing Nanocarrier

The spatial configuration of a 2D-ordered linear-chain carbon looks like a two-dimensionally distributed hexagonal set of the parallel carbon chains interconnected by the van der Waals forces. In this case, the distance between the carbon chains is estimated as 5Å [27,28].

The 2D-ordered linear-chain carbon can be considered also as a carbyne-enriched nanomatrix containing encapsulated oriented linear chains of carbon atoms—the monatomic carbon filaments.

Our understanding of the nanoarchitecture and the physicochemical properties of the 2D-ordered linear-chain carbon is still developing. For practical use of the carbyne-enriched nanomaterials, the ability to ensure the high stability of this nanomaterial is of key importance.

Carbyne, the “holy grail” of carbon allotropes, represents one of the examples of the unknown in science. The ideal one-dimensional form of carbon, an infinitely long linear chain of carbon atoms, named as a carbyne, attracted much interest due to its advanced mechanical and physicochemical properties, including the mechanical strength, predicted to be an order of magnitude higher than that of diamond [29,30,31]. Like a graphene, carbyne is just one atom thick, which gives it an extremely large surface area in relation to its mass. Carbyne was first described in 1885 by Adolf von Baeyer.

Back in the 1930s, astronomers discovered carbynes as one of the first molecules in interstellar space. Later, astronomers found signs of the presence of cosmic carbyne crystals in interstellar dust clouds. One of the possible routes for the formation of cosmic carbyne crystals from carbonaceous dust is connected with intensive synchrotron radiation under a coronal photon flux in the interplanetary medium. In this connection, carbyne crystals can be considered as out-of-this-world interstellar material. The unique space environment, along with the intensive electromagnetic and radiation fields and microgravity conditions, provides ideal conditions for the growth of the carbyne crystals that are more perfect than their counterparts grown on Earth. The Earth-grown carbyne crystals usually contain defects that induce instability in the crystal structure. Research has shown that crystal growth in microgravity conditions has a benefit due to the lack of buoyancy-induced convection, which affects the transport of molecules in the crystal.

The discovery of interstellar carbyne crystals can be considered as a key for understanding the mechanism of its formation and the environments in which its formation occurs, and also supports the concept of self-organization phenomena during the carbyne chain’s growth. The electronic configuration of a linear-chain carbon molecule contains two types of bonds: the (σ)-bond is responsible for the mechanical stability of the linear-chain carbon molecule, and the (π)-bond, along with mechanical stability, is responsible for the electrical properties, because the π-electrons are delocalized and hence belong to the whole chain of atoms. The electronic configuration of a part of a linear-chain carbon molecule is presented in Figure 1.

The linear carbon chains are observed in carbon vapor at temperatures above 5000 K. The technology for growing this unique carbon nanomaterial looks quite simple, since it demonstrates the ability to self-organize in a vacuum when condensed from carbon vapor at temperatures above 3150 K. However, pure carbyne in the condensed phase exhibits extreme instability due to its high chemical activity.

The growth of the macroscopic crystals of the carbyne is inhibited by the instability and high reactivity of this allotropic form of carbon. Carbyne physicochemical properties can be manipulated through the chain length, doping by nanoclusters and by a type of chain termination. Nanostructural stability depends on the linear carbon chain’s length. Free carbon-atom wires (CAWs) of any length must be terminated by molecular complexes to ensure their stability.

Most current research efforts are concentrated on searching for possibilities for the stabilization of the sp-hybridized carbon chains. The growth process determines the resulting properties of the carbyne-enriched nanomaterials.

During a long period, attempts to grow long carbyne molecules were limited by its extreme chemical instability. In 2016, a research team at the University of Vienna proposed and demonstrated a fundamentally new strategy to ensure the structural stability of extremely long sp-hybridized carbon chains containing more than 6000 carbon atoms through growing them within the long nanomatrix formed by double-walled carbon nanotubes [31]. The double-walled carbon nanotubes provided efficient structural stabilization of the encapsulated one-dimensional carbyne molecule, which is extremely important for designing the advanced functional nanostructural metamaterials. Such kinds of carbon nanotubes serve as nanoreactors and protect the sp-hybridized carbon chains from interaction with the environment. This outstanding result showed the new fundamental possibility of using the control of the nanomatrix spatial structure for programming the stability of the inserted sp-hybridized carbon chains.

The strategy, connected with the growth of the sp-hybridized carbon nanostructures in the composition of multicavity nanomatrices, seems to be the most promising way for the creation of the advanced carbon-based nanostructured metamaterials.

In 2019, an alternative modification of this strategy was proposed by creating a new class of carbon allotropes synthesized by encapsulating linear sp-hybridized carbon chains into cylindrical nanocavities formed within the sp-3 carbon hexagonal structure [32]. Because the sp-hybridized carbon chains are highly unsaturated, they tend to react to form additional bonds, resulting in the decay into both sp-2 and sp-3 structures. In this regard, it is possible to ensure the incorporation and isolation of sp-hybridized carbon chains inside some sufficiently wide and long nanocavity formed by sp-3-hybridized carbon to prevent the spontaneous reaction and decay of the sp-hybridized carbon chains. As such an insulating nanostructure, we chose a diamond hexagonal structure, which has a sufficiently extended nanocavity along the axis.

The result was the growth of an allotrope demonstrating the characteristic high-frequency vibrations associated with the sp-hybridized chain stretching modes and having a long time stability at room temperature environments.

Relatively recently, a research team in the Carbon Nanosystems Lab of the Physical Electronics Chair of the Physics Department of Moscow State University (Babaev, V.G., Guseva, M.B., Streletsky, O.A., Khvostov, V.V., Savchenko, N.F.) found new routes to encapsulate the oriented linear chains of carbon atoms—the monatomic carbon filaments into the matrix of amorphous carbon, thus creating bends and controlling the end groups in the process of ion-assisted pulse-plasma growth [27,28].

This is one of the most promising routes to obtain stable carbyne-enriched nanostructured-metamaterials, by growing them within the composition of the multicavity nanomatrix through the ion-assisted pulse-plasma deposition from the carbon plasma [28]. This technique also opens possibilities for the further increase of the long carbon chain stability through the assembling of atomic clusters of different chemical elements, for instance, silver, gold, titanium, etc.

Within a 2D-ordered linear-chain carbon nanomatrix, the carbon atom wires very weakly interact with each other (due to the van der Waals interaction), and therefore the properties of such nanomatrices are actually determined by the properties of individual CAWs. The ordered array of the one-dimensional carbon chains packed parallel to one another in hexagonal structures are oriented perpendicular to the substrate surface. In the carbon atom chains, each carbon atom is connected to the two nearest neighbors by the sp-1 bonds. The geometric characteristics of the 2D-ordered linear-chain carbon nanomatrix is presented in Figure 2.

Along the vertical axis (Y), the electrical conductivity of the nanomatrix is metallic. The charge transfer along the carbon atoms chain along the Y axis is carried out by delocalized electrons of the π-bond. Due to the absence of coupling between the carbon chains along the longitudinal and transverse axes (X, Z) (where the van der Waals forces only exist), the nanomatrix demonstrates dielectric properties along these axes. Such nanomatrices can be considered as an array of mutually-stabilizing CAWs, which also includes a set of different sp-phases that provides stabilization of the sp-hybridized carbon chains. The straight carbon chain becomes unstable as it lengthens. A carbon chain with self-forming kinks (see Figure 2) is a more stable system (energetically more favorable) than a straight carbon chain.

To ensure the stability of the growing carbon chains, it is necessary to provide a special requirement for the plasma-pulse-specific energy. The electron temperature of the plasma should not exceed the bond breaking energy in the carbon chains, since this leads to the “crosslinking” of these carbon chains and the formation of amorphous carbon with a short-range order of the diamond or graphite type.

The specific energy of the plasma pulse should exceed the breaking energy of the sp-2 bonds (614 kJ/mol) and the sp-3 bonds (348 kJ/mol) but should not exceed the breaking energy of the sp-1 bonds (839 kJ/mol) in the evaporating carbon chains. The controlled bond-breaking and sp-phases transformation can be provided through the predictive ion-assisted stimulation with specific energy levels.

At large thicknesses of the growing carbyne-enriched nanomatrix, the probability of interchain interaction and the formation of crosslinks between the carbon chains increases. In this regard, the growing nanomatrix is stimulated by the argon ions and, with an increase in thickness, its structure is additionally stabilized by the hydrogen ions injected into the arc discharge plasma during the carbon condensation process [28].

The 2D-ordered linear-chain carbon nanomatrix represents a multicavity nanostructure containing vacant functional nanocavities available for the assembling by atom clusters of various chemical elements. The schematic representation of the vacant functional nanocavity of the nanomatrix available for nanocluster assembling is presented in Figure 3.

By the cluster-assembling of the spatial structure of the 2D-ordered linear-chain carbon nanomatrix with various molecules and specific catalytic agents and chemical elements, the properties of the nanomatrix can be modified or new structural and physicochemical properties can be added.

Cluster-assembly of the nanomatrix can occur both without chemical interaction (so called intercalation) and with the rupture of the π bonds, which can lead to an additional reaction. For instance, by assembling the 2D-ordered linear-chain carbon nanomatrix with calcium clusters, which suck up hydrogen molecules, a high-density, reversible hydrogen storage device is created.

In accordance with the geometric characteristics of the 2D-ordered linear-chain carbon-based nanomatrix and the configuration of the vacant functional nanocavity available for nanocluster assembling, as shown in Figure 2 and Figure 3, the scheme of the multiple heteroatom doping functionalization of the nanocarrier, based on the multicavity nanomatrix, is presented in Figure 4.

The 2D-ordered linear chain carbon nanomatrix could serve as an efficient basis for the design and the growth of the new carbon-based nanostructured metamaterials with unique electrophysical, optical, structural, topographic and chemical properties. The spatial structure of such a nanomatrix can also self-adjust to the structure of the embedded atom clusters.

With a small chemical doping, the carbyne-enriched functionalizing nanocarriers can be transformed into a controllable piezoelectric material. Assembling the carbyne-enriched functionalizing nanocarriers by the piezoelectric nanomaterials clusters, for instance, by lithium atoms or zinc oxide (ZnO) nanoparticles, can transform them into the piezoelectric nanogenerators that can be used to control the electric charge distribution within the nanomatrix growing zone. This effect can also be reversed—applying an electric field to a piezoelectric nanogenerator will cause it to change shape or deform. The deformation generated in the piezoelectric nanogenerators is proportional to the magnitude of the applied electric field.

### 3.2. Pulse-Plasma Deposition Reactor for Growing the Functionalizing Nanocarriers

The cathodic arc plasma deposition (CAPD) or arc-PVD (PVD is physical vapor deposition) is a physical vapor deposition technique in which an electric arc is used to vaporize material from a cathode target. The vaporized material then condenses on a substrate, forming a 2D-ordered linear-chain carbon nanomatrix. The experimental set-up for the ion-stimulated pulse-plasma deposition of the 2D-ordered linear-chain carbon nanomatrix with the capability of cluster-assembly by various chemical elements is presented in Figure 5.

The main components of the pulse-plasma deposition reactor for growing the 2D-ordered linear-chain carbon nanomatrix are as follows: the vacuum chamber; pulse-plasma carbon generator; the ion source for ionic stimulation; the target assembly with removable target material. The ion and plasma beams intersect above and at the substrate surface. The ion beam irradiation of the deposition zone forms bends in the attached carbon chains which stabilize the growing chain ensemble.

The evaporation of the carbon plasma sheaf from the main discharge graphite cathode 3 is caused by the local heating of the graphite surface by electron bombardment to T = 3000 C. The chains of carbon atoms, C_n_ (where n = 1, 2, 3, …), formed in the plasma sheaf, are directed by electrode system to impinge upon the surface of the substrate where the polycondensation of the carbon chains takes place.

The schematic representation of the pulse-plasma carbon generator installed in the reactor of the experimental set-up (Figure 5) is shown in Figure 6.

An arc discharge is ignited between the main discharge of cathode 1 and the main discharge anode 2 (which are preferably separated by a voltage of about 200 V) by means of auxiliary discharge between the ignition anode 6, the main discharge cathode 1 and the auxiliary discharge anode 4 surrounding the main discharge cathode 1. The auxiliary discharge is ignited by means of ignition cathode 5.

The main discharge cathode (item 1 in Figure 6) design depends on the purposes of deposition and can be manufactured as a composite structure, containing cylindrical rods from various materials, used for heteroatom doping, for instance, with silver, tungsten, gold, etc. An example of the design of a cylindrical main discharge cathode is presented in Figure 7.

The capacitors C1 and C2 shown in the diagram are connected with a power source that supplies an alternating voltage (100–300 V). In this case, a pulsed voltage with an amplitude of 800 V is applied to the ignition electrodes. The inductance L shown in the diagram is used to reduce the rate of the current increase to the required value. The growth of the carbon nanomatrix is stimulated by the irradiation with Ar+ ions. The flow of Ar^+^ ions is formed using a source of low-pressure ions, which is installed in a separate section of the reactor vacuum chamber (see Figure 5).

The energy of ions bombarding the substrate surface depends on the substrate bias voltage, being varied in the range 0–300 eV by both the carbon plasma parameters and the ion source extractor voltage, depending on the parameters of the ion-stimulated pulse-plasma deposition. The nanomatrix can be deposited onto Si wafer, various metals and NaCl single crystals at an ion energy of 150 eV. Before deposition, the reactor chamber was pumped down to a residual pressure of 10^−4^ Pa. The operating pressure during the deposition was 10^−4^ Pa.

The structure of the bonds in the grown carbon-based nanomatrices can be programmed by the processes of self-organization and autosynchronization of the growing nanostructures. A number of experimental studies show that the sp-bonds and the sp-hybridized carbon nanostructures are formed only in a narrow range of optimal ion-stimulated pulse-plasma deposition parameters and modes [33].

The thickness of the grown nanomatrix is programmed by the number of pulses, by the energy per pulse, by the capacitance of the main discharge capacitors and by the charging voltage of these capacitors. The minimum coating thickness is 0.1–0.5 nanometers. With the deposition frequency of 1–5 Hz, the temperature of the samples does not exceed 60–80 °C.

The ion-assistance during the nanomatrix pulse-plasma growth has a significant influence on its structural and physicochemical properties. Transmission electron microscopy study of the samples show that the ion-assistance energy level significantly affects the surface nanopattern shapes, sizes and spatial localizations. The specific conductivity of ion-assisted samples is 10^3^–10^4^ times larger than the conductivity of the samples deposited without ion assistance.

## 4. Vibration-Assisted Activation of the Pulse-Plasma Growth Zone

### 4.1. Ordered Pattern Formation at the Nanoscale

Structural self-organization and ordered pattern formation are the universal and key phenomena observed during the growth and cluster-assembly of the 2D-ordered linear-chain carbon-based nanomatrix at the ion-stimulated pulse-plasma deposition [34,35]. Self-organization phenomena are observed also during the cluster-assembling of the nanocavities by atoms of various chemical elements due to formation of new chemical, interatomic and intermolecular bonds. The pulse-plasma deposition zone is a vibration-sensitive media for which the universal laws of cymatics are valid [36].

The pattern’s excitation phenomena in the pulse-plasma deposition zone are programmed by the interaction of several competing mechanisms, in particular through the thermoelectric convection excitation, by the state of stress in the deposited nanomatrix and by the self-synchronization of the self-excited oscillatory cells in the deposition region.

The self-organization and formation of surface patterns are most pronounced when the grown system is supplied with extra energy and by nanosized active centers.

Transmission electron microscopy has demonstrated that the structure of a nanomatrix grown without ion-assistance is homogeneous, while the structure of nanomatrices grown with ion-assistance becomes inhomogeneous [37]. For the case of doping a 2D-ordered linear-chain carbon nanomatrix with silver clusters, transmission electron microscopy has shown that, with an increase in the energy flux into the growing region, the average size of the active nucleation centers decreased with a simultaneous increase of their quantity [38].

Recent research demonstrates the 2D-ordered linear-chain carbon-based nanostructures that grow from high-temperature carbon plasma occur in accordance with the unified templates, the nanoarchitectures of which are determined by the vibrational state of carbon plasma molecules.

### 4.2. Model Experimental Systems for Vibrational Activation

A set of model experimental systems demonstrate how vibrations provide significant influence on the structures of the deposited model nanomaterial. Paper [39] demonstrated a new approach of the evaporated material vibration-assisted thermal deposition in vacuum chambers.

Earlier, papers [40,41] were devoted to atomic deposition experiments, wherein the authors studied the nanoscale self-organization phenomena on the substrate surface using standing surface acoustic waves (SAW).

Molecular dynamics simulations were used to describe the structuration of the physical driving mechanism. However, this research does not take into account the vacuum conditions influenced during the deposition processes. Acoustic waves cannot propagate in the low-pressure environment. At the same time, the acoustic waves can be transmitted from the vibration source to the solid as mechanical vibrations with a specific frequency. In the result, the substrate mechanical vibrations during the deposition could influence the surface morphology and the structure of the deposited nanomaterial.

Selenium, having a variety of allotropic phases, is a convenient material for the fundamental research of the mechanical vibrational influence during the nanostructure’s deposition. Tellurium, as well as hexagonal selenium, is a typical crystalline semiconductor whose atoms form polymeric, covalently bonded helical chains packed into a hexagonal lattice through the van der Waals forces. Due to this specific nature of tellurium, during the nanostructure deposition the formation of the nanowires, nanotubes, nanorods, etc., are observed. Such kinds of specific nanostructures can be considered as the model experimental systems that can demonstrate fundamental phenomena observed at the vibration-assisted thermal deposition of the evaporated material in a vacuum with acoustic wave frequencies.

During the nanostructure deposition, the mechanical oscillations were applied to the substrate with input frequencies of 0, 50, 150 and 4 kHz at the deposition rate of 0.3 nm/s, and the vacuum chamber pressure of 7 × 10^−3^ Pa. As can be seen from the atomic force microscopy (AFM) images of 150 nm thick tellurium nanostructures, presented in paper [39], the acoustic waves applied to the substrate resulted in morphological changes, demonstrating the self-organization of the nanostructures.

As was demonstrated by the model experimental systems, vibrational activation is capable of transforming the orientation of the nanostructures in the grown nanomatrix.

In the late 18th century, German physicist Ernst Chladni demonstrated the organizing power of sound and vibration in a visually striking manner. In the 1950s, the study of wave phenomena was continued by Swiss scientist and anthroposophist Hans Jenny, who named the research field as “Cymatics” (“kyma” is the Greek word for wave) [36].

Under this term, he summarized all phenomena which appear when vibration and sound meet substance. Sound is both a wave and a geometric pattern at the same time. Hans Jenny also discovered that higher frequencies produced more complex shapes. As the frequency increases, the disappearance of one pattern may be followed by a short chaotic phase before a new, more complicated and stable structure appears. As the amplitude increases, the movements become more and more rapid and violent, sometimes with small eruptions. The forms, shapes and patterns of motion that emerged turned out to be primarily a function of frequency, amplitude and the inherent characteristics of the various materials. A significant detail in Dr. Jenny’s research of vibrational forms in fluids and gases is that, after the first disturbance activation in a fluid, gas or in a flame, this medium becomes sensitive to the influence of sound or vibrations.

The acoustic hologram generated in the nanostructure growth zone through external vibrational–acoustic activation is capable of controlling the growth process, cluster-assembling and formation of chemical bonds.

In accordance with the universal laws of cymatics and the unified template (Mereon Matrix) approach, during the vibration-assisted activation of the pulse-plasma deposition zone, the excitation of the self-organized patterns occurred in accordance with the three-dimensional unified template [42,43]. The Mereon Matrix is a three-dimensional template of a dynamic geometric process. The structure of the material systems is formed on the basis of the unified templates, the forms of which are defined by presence of vibrations in the system. The connection between shape and vibration are defined through the Mereon Matrix. Accordingly, for the case we are considering, the formation of the 2D-ordered linear-chain carbon-based nanomatrix structure occurs in accordance with the unified template patterns described by the cymatics laws. The unique structural and physicochemical properties of the nanostructured metamaterials arise not from the properties of the forming initial materials, but from the specific design of their arrangement, geometry and orientation.

The geometrical structure of the metamaterials usually includes a repeated pattern at a scale that is smaller than the wavelengths of the phenomena they influence. Nanostructured metamaterials demonstrate unique physicochemical properties determined by their geometrical structure.

### 4.3. Surface Acoustic Waves-Assisted Micro/Nanomanipulation

Assuming that the 2D-ordered linear-chain carbon-based functionalizing nanocarriers are acoustically sensitive nanomaterials [27], we propose the predictive tailoring of the nanoarchitecture, patterning, vibrational characteristics and multifunctionality of the nanocarriers using the surface acoustic wave (SAW)-based toolkit. In particular, we propose to use the technology of the nanomaterials growing onto acoustically excited piezoelectric active substrates.

Assisting the nanomaterial growth through generating the Rayleigh-type SAW leads to patterning phenomena characterized by substantial lateral changes in the nanostructures, thicknesses and properties [44]. Certain frequencies of acoustic vibrations are capable of forming various geometric shapes. In addition, changes in the crystal structure are also induced. There exist two basic types of the bulk acoustic waves. The first one is the longitudinal wave, in which the oscillations of the particles are only in the direction of the wave propagation. The second one is the shear wave, in which the particle displacements are orthogonal to the wave propagation. The size and distribution of pattern formation can be controlled through the adjustment of deposition parameters and the SAW properties.

Changing the acoustic driving frequency can be employed to modify the nanopattern size. The use of combinations of vibrations in different frequency ranges makes it possible to purposefully control the nanostructure of the grown metamaterial.

The piezoelectric elements and layers are also capable of generating electromagnetic emissions [44]. Acoustic and electromagnetic holograms with specified frequencies and spatial characteristics are capable of providing the spatial markings of the structure of the grown 2D-ordered linear-chain carbon-based nanomatrix.

We propose to apply the new synergistic effect through simultaneous vibration-assisted self-organized wave pattern excitation along with the manipulation of their structural and physicochemical properties through the electromagnetic field. The interaction between the inhomogeneous distribution of the electric field generated on the vibrating piezoelectric layer and plasma ions will serve as an additional energizing factor that controls the local pattern excitation as well as the self-organization of the nanostructures.

Application of the inverse piezoelectric effect during the 2D-ordered linear-chain carbon-based nanomatrix ion-assisted pulse-plasma deposition can be provided through the deposited piezoelectric layer (see Figure 8).

The piezoelectric layer converts the supplied electric energy signals into the mechanical acoustic oscillations that propagate through the substrate and growth of the 2D-ordered linear-chain carbon-based nanomatrix. Applying an alternating current or radio frequency excitation to the electrodes deposited onto the piezoelectric material generates an acoustic wave that propagates in the direction perpendicular to the surface of the deposited nanomatrix into the bulk medium (bulk acoustic wave) or along the surface of the growing nanomatrix (SAW).

Using different acoustic excitation frequencies and waveforms generated by piezoelectric elements excites the specific unified templates that mark the spatial geometry for growing the nanostructures. Such vibration–acoustic activation can be used to program the required nanoarchitecture of grown carbyne-enriched nanomatrices. The simultaneous use of direct and inverse piezoelectric effects opens up the possibility of providing the interactive ion-assisted pulse-plasma growth of the 2D-ordered linear-chain carbon-based nanomatrix (see Figure 9).

Using direct and inverse piezoelectric effects in a nanostructure monitoring system, one piezoelectric transducer is electrically excited, causing a vibration which can be detected by a multitude of piezoelectric transducers operated in the sensor mode. By evaluating and comparing the excitation signal and the output signals of various sensors, it is possible to obtain information regarding the current state of the growth of the nanostructure.

Our research performed for model experimental systems proves SAW-assisting the ion-plasma growing the 2D-ordered linear-chain carbon-based nanomatrix leads to cymatic patterning phenomena characterized by significant changes in the matrix nanoarchitecture as well as to the chemical bond transformation and nanocarbon structural phase transformation.

In the model experimental system under study, the complex synergistic effect is realized through the merging of several physicochemical processes: the interaction between the pulsed flow of high-temperature plasma and the ion flow, and the carbon chains self-organizing phenomenon during condensation from the carbon vapor in a vacuum accompanied by growing nanolayers activation by the SAWs.

In our model experimental system to excite the SAWs, we used the lithium niobate (LiNbO_3_) piezoelectric substrates with a distributed electrode system as well as piezoelectric ceramic elements with a circle electrode system. These SAWs are like seismic waves excited by earthquakes, and their stress is highly concentrated near the surface. The advantage here is that the transducer on the piezoelectric substrate electrically generates the nanoquakes on demand, launching spatially and temporally tailored waves of a controlled magnitude. In particular, we also provide excitation of the standing SAWs of the Rayleigh type on a LiNbO_3_ substrate. The research was conducted at a SAW excitation under different ranges, including the sonic and ultrasonic range (4–50 kHz). Our studies have shown that the growth rate of a 2D-ordered linear-chain carbon-based nanomatrix significantly increases with vibration rather than in the absence of vibration. The nanomatrix growth rate with ultrasonic vibrations increases with the frequency increase up to a certain value, and after this frequency value, the growth rate remains almost constant.

The nanomatrix surface morphology, grown with the assistance of SAWs, and investigated with the cross-section scanning electron microscopy (SEM) analysis, is observed as more compact and smoother in comparison with the nanomatrix grown without the assistance of SAWs.

The sp-phase transformation in the nanomatrix growing zone during vibrational-acoustic manipulation through the SAWs is associated with the bonds breaking at the input high-frequency vibrational energy exceeding the sp-2 and sp-3 bonds breaking energies. We have used the SAWs for manipulating the phase transformations during the growth of mixed-nanocarbon structures, where the sp, sp2 and sp3 bonds occur.

The SAWs of the ultrasonic range have the capability of influencing the formation of the chemical bonds during the 2D-ordered linear-chain carbon-based nanomatrix ion-assisted pulse-plasma growth and are capable of breaking the sp carbon chemical bonds as well as the long chains of carbon. In particular, it was found that oscillations of the low-frequency sound range stimulate the formation of the sp2 and sp3 phases (the diamond/graphite phases). With a shift to the high-frequency, ultrasonic region, the formation of the sp phase is stimulated. An example of the binding energy ratio for carbyne (sp) and graphene-like (sp2) hybridized bonds, obtained using X-ray photoelectron spectroscopy (XPS), is shown in Figure 10.

## 5. Raman Spectrum-Based Vibrational Signature

The 2D-ordered linear-chain carbon-based functionalizing nanocarriers can be considered as both acoustic and electromagnetically sensitive nanostructured metamaterial [27]. The vibration of sp-hybridized carbon chains encapsulated into the multicavity nanomatrix occurs due to the van der Waals interactions between them. Each 2D-ordered linear-chain carbon-based nanomatrix grown can be characterized by the unique vibration signature formed by a set of individual CAWs [45]. The sp-hybridized carbon chains oscillate like elastic strings. Like the tuning of a guitar string, this vibration behavior can be determined based on length and tension (Figure 11).

The analysis of a 2D-ordered linear-chain carbon-based nanomatrix that can be performed with Raman spectroscopy goes beyond just simple chemical classification and refers to the category of vibrational spectroscopy [46,47,48,49]. The Raman spectroscopy analyzes a sample through the molecular vibration excitation and the subsequent decrypting of this vibrational interaction. The Raman spectrum of a multicavity nanomatrix will contain a series of peaks that correspond to the different vibrational modes in a molecule. The position and linewidth of these peaks can be so unique to a system that they are considered a fingerprint for the molecular species.

The Raman vibrational spectrum contains a unique vibrational signature of the scattering molecule with a high resolution and can be used for precision identifying molecules. In this regard, Raman spectroscopy is considered as a key analytical characterization technique for the deep study of carbon-based nanomaterials and nanoscale surfaces [46,47,48,49,50].

Raman spectroscopy can be used as an effective tool for determining the structural arrangement that characterize two different forms of the same type of carbon-based nanomaterial, to distinguish between different allotropes of the same type carbon nanomaterial, to identify the phases and phase transitions, to determine which regions of a nanomaterial are amorphous or crystalline, to identify whether there are any defects present within the nanomaterial and to determine the shape of the nanomaterials.

The Raman spectroscopic technique, as a promising self-sufficient technique, has enabled unprecedented insight into the physicochemical and structural properties of the carbon-based nanomaterials, and especially of low-dimensional nanocarbon allotropes. In particular, Raman spectroscopy is a key technique applied for bond structure characterization in nanocarbon allotropes. For instance, the spectral region between 1800–2400 cm^−1^ associated with sp-hybridized carbon (the carbyne phase).

In the process of the 2D-ordered linear-chain carbon-based nanomatrix growth, its nanoarchitecture can be modified using the SAWs generated in a certain frequency range.

The SAWs are also can be represented as a nanoquakes, which are capable of modifying the 2D-ordered linear-chain carbon-based nanomatrix phonons, its Raman response, and therefore can be used for fine-tuning the vibrational properties of the growing nanomatrix. Changing the modes and parameters of the ion-assisted pulse-plasma growth of a 2D-ordered linear-chain carbon-based nanomatrix allows for fine-tuning its vibrational signature.

By analogy with the proposed earlier “genetic barcode” and “molecular barcode” concepts, in 2021, the new concept of a universal “Raman Barcode” was proposed, obtained by converting the sequences of bands in the Raman spectrum for the accurate express recognition of various combinations of molecules [50,51].

The Raman spectra contain hidden quantum information that can be extracted and recognized using the sonification technique into the acoustic holograms [52]. Like in the science of stellar sound waves, the molecular vibration frequencies are inaudible to the human ear. The understanding of these molecular vibration frequencies represents a revolution in nanomaterials science. The sonification of the 2D-ordered linear-chain carbon-based nanomatrix vibrational signatures open the possibilities to transform them into acoustic holograms, which significantly expands the research opportunities.

## 6. Data-Driven Tailoring of Architecture and Functionality of Nanocarriers

Nowadays, research on nanomaterials science is rapidly entering the phase of a data-driven age. For the predictive growth of the 2D-ordered linear-chain carbon-based functionalizing nanocarriers with a unique set of programmable microarchitectures and physicochemical properties, by using the extensive experimental testing, we propose to apply a new paradigm in materials science—a science based on data and deep materials informatics [53,54,55]. In this research area, the experimental data is a new resource, and knowledge is extracted from the datasets of materials.

During recent years, the application of data science techniques for nanomaterials research and development has demonstrated significant achievements due to their outstanding capabilities to effectively extract the critically-important data-driven linkages from various nanomaterial input representative data to their output nanoarchitectures and physicochemical properties [56,57].

Applying a deep materials informatics approach opens up unprecedented possibilities for the predictive programming of the spatial structure of the grown 2D-ordered linear-chain carbon-based nanomatrix at a fundamentally new level.

We propose the fine-tuning of the vibrational signature, heteroatom doping, functionality and nanoarchitecture of 2D-ordered linear-chain carbon-based functionalizing nanocarriers by using precision SAW-assisted manipulations by the pulse-plasma growth zone, combined with the data-driven carbon nanomaterials genome approach. This approach represents a digital toolkit based on deep materials informatics belonging to the fourth scientific paradigm.

The proposed data-driven carbon nanomaterials genome approach establishes link-ages between the key modes and parameters of the ion-assisted pulse-plasma growth of the functionalizing nanocarriers and their resulting nanoarchitectures and physicochemical properties through a set of multifactorial computational models developed with the use of extensive experimental data for the selected set of key descriptors or fingerprints.

A critical and key condition for the data-driven carbon nanomaterials genome approach development is the correct selection of the set of descriptors or features for inclusion into the multifactorial computational models that characterize the nanomaterials under study, which have a well-known correlation between the target and other properties. The descriptors can be classified as numeric or categorical.

Based on the use of the formalized universal linkages, it becomes possible to predict changes in target nanoarchitectures and physicochemical characteristics for various ion-assisted pulse-plasma synthesis conditions, and vice versa, to predict the synthesis technological modes based on the required structural and physicochemical characteristics of the carbyne-enriched nanomatrix.

A set of multifactorial computational models mentioned above can be developed by using the modern data mining methods (feedforward neural networks, deep learning neural networks, multiple adaptive regression splines, etc.). Application of the artificial neural networks toolkit opens unique possibilities to identify and describe, through the multifactorial computational models, all the hidden linkages of the carbyne-enriched nanomatrix vibrational signatures with their ion-assisted pulse-plasma synthesis modes and target physicochemical properties and nanoarchitectures.

The artificial neural network (ANN) is a machine learning (ML) methodology currently used for predictive modeling in many research areas. A detailed description of the ANN methodology can be found in a number of papers, e.g., [56,58,59,60,61,62].

The most modern approach to building multifactorial models based on the use of deep learning neural networks (deep learning), which allows to confidently extrapolate the identified patterns and solve forecasting problems, is implemented in the unique data science software platform PolyAnalyst, developed by Megaputer Intelligence [63,64]. The data science platform PolyAnalyst is the industry standard for extracting usable knowledge from large amounts of structured and unstructured data [64].

We propose to consider the Raman spectra-based molecular fingerprints, the vibrational signatures, as one of the key descriptors for incorporating into the data-driven carbon nanomaterials genome approach because this descriptor can catch the main peculiarities of the carbon-based nanomaterials. In this case, the multifactorial computational models will be developed using a set of Raman vibrational spectra of the investigated 2D-ordered linear-chain carbon-based functionalizing nanocarriers along with the key modes and parameters of the ion-assisted pulse-plasma synthesis.

In this case, the multifactorial computational models are considered as carriers of information regarding the linkages between the 2D-ordered linear-chain carbon-based nanomatrix vibrational characteristics and the modes and parameters of the ion-assisted pulse-plasma growing.

Compared to existing computational approaches such as density functional theory (DFT), used for modeling carbon-based nanomaterials properties, deep materials informatics allows much faster prediction of properties and opens up the possibility of describing disordered structures. This circumstance is of decisive importance for the discovery of new carbon-based nanomaterials which would not have been obvious with intuition alone.

The utility and versatility of examples of the artificial neural network application, one of the most promising methods of data science for predicting certain macroscopic properties of energetic compounds, based on a trained set consisting of a large data set, are presented in the following references: [56,57]. This approach demonstrates similar predictive accuracy in comparison with the similar data derived from well-known empirical models.

The proposed schemes of tracking and tailoring the key descriptors and linkages for incorporating into the data-driven carbon nanomaterials genome approach are presented in Figure 12.

## 7. Electromagnetic and Acoustic Activation of the Functionalizing Nanocarriers

The proposed modifications of the production technology based on the transformative energetics concept open up new prospects for the further enhancement of the energy resources and controllability of the EM reaction zones.

The 2D-ordered linear-chain carbon-based functionalizing nanocarriers in this case can serve as transformers of the acoustic hologram into the electromagnetic emission, as well as directed self-assembly agents in an external electromagnetic field to manipulate the energy release domain localizations in the EM reaction zones.

For the predictive manipulation of the spatial distribution of the induction and energy release domains in the EM reaction zones, we propose the activation of the above-mentioned nanocarriers with the use of the earlier proposed plasma-acoustic coupling technique [65], as well as with Teslaphoresis force field [66]. This activation includes the directed self-assembly of the above-mentioned nanocarbon-based additives and functionalizing nanocarriers.

As has been shown recently, the predictive manipulation by self-organized wave pattern excitation and by micro- and nanoscale oscillatory networks are one of the most effective ways to access the properties of the EM and solid propellant reaction zones [65,67]. Programmed acoustic emissions from the oscillated plasma arc disc into the EM reaction zones opens possibilities for manipulation by the networks of the micro- and nanostructures. Application of different acoustic frequencies emitted by the plasma arc emitter will activate specific properties in the EM reaction zones and will change the special localization and properties of the EM reaction zones. Moreover, such acoustic emissions are capable of exciting the self-organization of the EM reaction zones.

The plasma-acoustic coupling mechanism transforms the input electrical energy into the directed acoustic energy. The plasma glow discharge, corona discharge or electric arc then acts as a massless radiating element. It creates compression waves in the gas. At the same time, the plasma arcs have zero weight. Generation of the plasma arc within a magnetic field perpendicular to its current path results in a Lorentz force on the charged particles, causing the arc to sweep about the center of the coax, forming a plasma disc. The technology of plasma arc emitters, modulated by acoustic frequencies, can provide manipulation by self-organization and by the self-synchronization of the micro- and nanoscale oscillatory networks and the self-organized wave pattern formation in the EM reactionary zones. For instance, the plasma arc force field emitter can be installed over the burning surface of the solid propellant end-burning charge with the special design of the electrode system.

Recent experimental studies have confirmed that, during an arc discharge, the vibrating carbon nanotube (CNT) is capable of generating an electromagnetic field [68]. Doping the 2D-ordered linear-chain functionalizing carbon-based nanocarriers with clusters of piezoelectric nanomaterials turns them into the piezoelectric nanogenerators that can convert a plasma-excited acoustic hologram into the electromagnetic emission. This electromagnetic emission can then be used for manipulating the micro- and nanoscale oscillatory networks in EM reaction zones. Transformation of an acoustic hologram into an electromagnetic hologram through the use of functionalizing nanocarriers doped with clusters of piezoelectric nanomaterials is schematically shown in Figure 13.

A high-frequency electromagnetic field applied through the plasma arc emitters over the EM reaction zones is capable of inducing the directed self-assembly of the carbon-based catalytic nanoadditives together with the 2D-ordered linear-chain carbon-based functionalizing nanocarriers. Such self-assembly is capable of changing the spatial distribution of the induction and energy release domains in the EM reaction zones.

Relatively recent research conducted at Rice University experimentally proved a new phenomenon that opened the way for the direct self-assembly of the low-dimensional carbon allotropes. According to this experimental study, a set of carbon allotropes, as well as other nanomaterials, can assemble themselves at a large distance depending on the level of the used high-frequency electromagnetic emission [66].

The traditional directed self-assembly of the nanomaterials using electric fields have been limited to small scale structures, but with the Teslaphoresis force field (the electrokinetic phenomenon), this limitation has been lifted. For example, the CNTs, when exposed to a Teslaphoresis force field, demonstrate polarization and self-assembly into long parallel arrays at the macro level and can serve as electric current conductors.

Application of the plasma-acoustic coupling-based technique as well as the Teslaphoresis force field-driving nanocarrier-directed self-assembly in the EM reaction zones is opening new ways for the inertial-free control by the structure and properties of the reaction zones with minimum expenses of energy and opens new doors for producing extremely small thrust impulses for the extra-precise attitude control of deep-space-capable small satellites.

## 8. Outlook and Perspectives

The next generation of aerospace propulsion systems requires energetic and propulsive materials to further increase the stored potential energy and thermodynamic performance. In this regard, the predictive programming of the storage and release of high-density energies on the required time scales to ensure the required efficiency is of great importance. Current research trends in the area of nanoenergetic materials are oriented on performance and safety enhancements through the application of a variety of catalytic nanoadditives and the development of environmentally-friendly EMs.

Recently proposed transformative energetics concepts, based on the predictive synthesis at the nanoscale, can be considered as a pathway towards the development of the next generation of nanoenergetic materials.

Within the transformative energetics concept, we propose a set of new approaches and technological solutions that will significantly expand its capabilities and versatility for the development and synthesis of promising EMs and solid propellants.

Our research has also provided a new strategy to develop the new generation of green high-energy ePropellants.

An important feature of the 2D-ordered linear-chain carbon-based functionalizing nanocarriers is the ability to fine-tune their nanoarchitectures and physicochemical properties. However, without using the deep materials informatics approach, such tuning through trial and error is extremely difficult.

Application of various acoustic exciting frequencies and waveforms, generated in the 2D-ordered linear-chain carbon-based functionalizing nanocarriers growth zone, excites and creates specific unified templates for the growth of the nanostructures, and can be used for programming the required nanoarchitectures of the grown functionalizing nanocarriers.

The use of the SAW toolkit makes precise manipulation possible by bond breaking and formation during the ion-assisted pulse-plasma growth, and, accordingly, by sp-phase transformations in the nanomatrix growth zone. The possibility of programmable Chladni pattern excitation at the nanoscale is a key approach to controlling the nanoarchitectures and properties of the grown 2D-ordered linear-chain carbon-based nanomatrices. Since the required combination of sp-phases in the composition of the nanomatrices being grown can be provided in a narrow range of technological growth parameters, the selection and exact provision of which is a difficult task, the use of the SAW toolkit opens up new possibilities for the precision tuning of parameters in the nanomatrix growth zone to provide the required combination of the sp-phases.

The 2D-ordered linear-chain carbon-based functionalizing nanocarriers, assembled by the piezoelectric nanomaterial clusters, can also serve as the acoustic radiation converters into electromagnetic emission, and vice versa. Acoustic and electromagnetic holograms with specified frequencies and spatial characteristics are capable of providing the spatial markings of the structure of the grown 2D-ordered linear-chain carbon-based functionalizing nanocarriers.

The combined use of the SAW toolkit, heteroatom doping and the data-driven carbon nanomaterials genome approach at the functionalizing of the nanocarrier ion-assisted pulse-plasma growth creates a synergy effect that makes it possible to multiply the efficiency of the used approaches. The vibrational signature of the functionalizing nanocarriers can be programmed in several ways—through the fine tuning of the energy and the repetition rate of the plasma pulses and ion stimulation energy, through heteroatomic doping with clusters of atoms of various chemical elements, by programming the composition of the main discharge cathode and the distribution of cathode spots and by the activation of the nanomatrix growing zone by the SAW.

Taking into account above-described approaches, the modified concept of the TEM multistage synthesis is schematically presented in Figure 14.

Application of the proposed new approaches and technological solutions opens new possibilities for smart control by the excited-state of the EM reaction zones and by the self-organized wave patterns excitation to access the properties of the EM reaction zones: the micro- and mesoscale and spatial distributions of the induction and energy release domains.

The proposed application the plasma-acoustic coupling-based technique, as well as the Teslaphoresis force field-driving nanocarriers-directed self-assembly in the EM reaction zones, is opening new ways for the inertial-free control by the structural and physicochemical properties of the reaction zones with minimum expenses of energy, and which opens new doors for producing extremely small thrust impulses for the extra-precise attitude control of deep-space-capable small satellites.

In summary, we have proposed several concepts and technological approaches that are capable of unlocking new potential opportunities for the predictive extracting of extra energy from nanoenergetic material systems at the nanoscale through the synergistic use heteroatom doping, the SAW-based tool-kit and the deep materials informatics-based toolkit of the 2D-ordered linear-chain carbon-based functionalizing nanocarriers ion-assisted pulse-plasma growing. As extra energy-extracting means from transformative energetic materials, we propose the use of the plasma arc force field emitter-based technique and Teslaphoresis force field-based technique.

The novelty of this research work is that we have proposed new pathways for the next generation of high-end nanoenergetic materials performance and safety enhancements that can be used for the development of the future of multimode solid propulsion systems and deep-space-capable small satellites.

## Figures and Tables

**Figure 1 nanomaterials-12-01041-f001:**
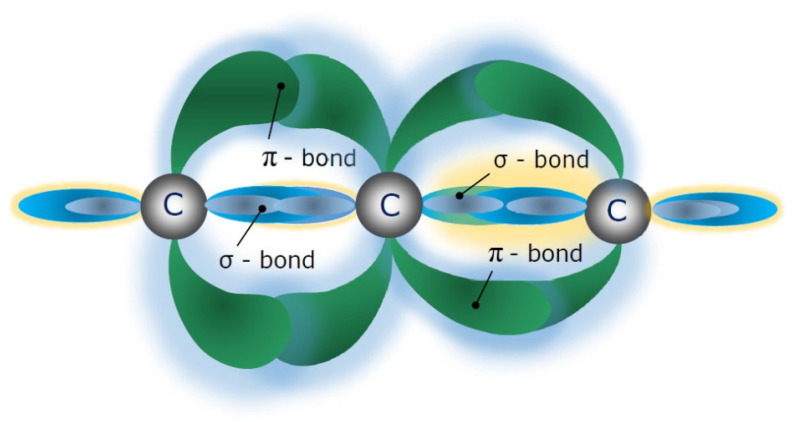
The electronic configuration of a fragment of a linear-chain carbon molecule.

**Figure 2 nanomaterials-12-01041-f002:**
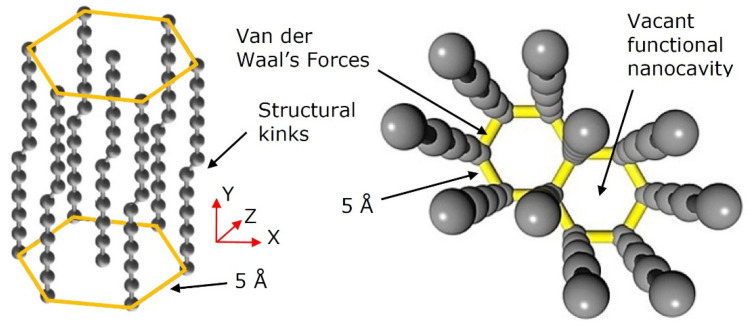
Spatial structure of the 2D-ordered linear-chain carbon-based nanomatrix fragment, containing the nanocavity, available for heteroatom doping.

**Figure 3 nanomaterials-12-01041-f003:**
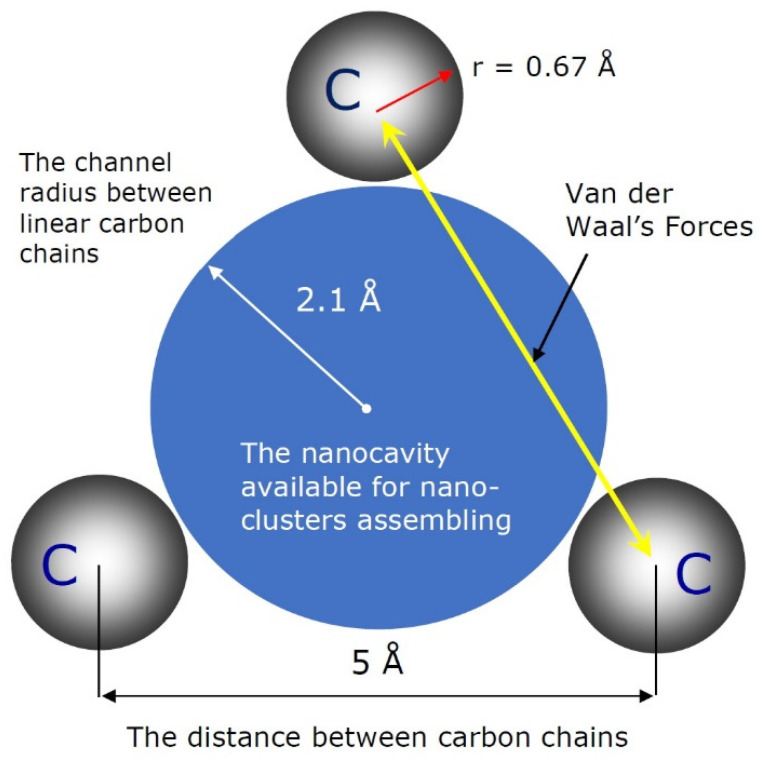
Schematic representation of the vacant nanocavity of the 2D-ordered linear-chain carbon-based nanomatrix available for heteroatom doping.

**Figure 4 nanomaterials-12-01041-f004:**
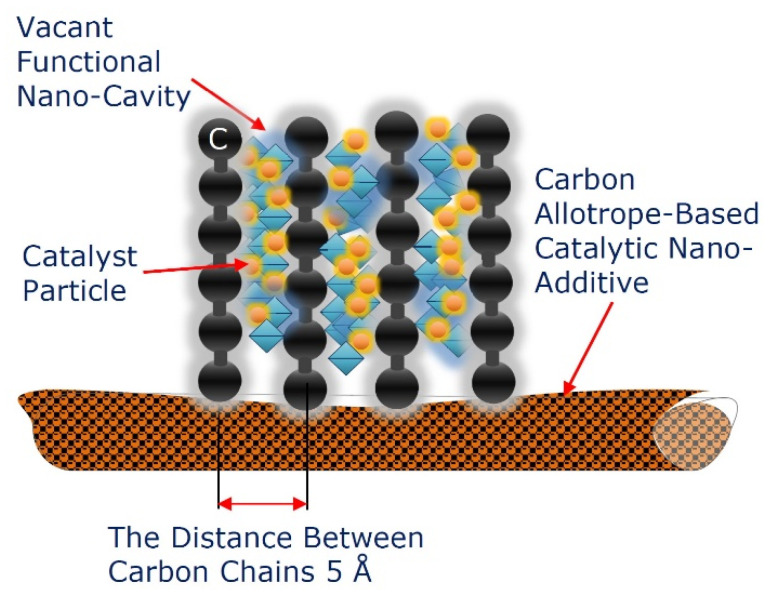
Schematic representation of the multiple heteroatom-doped functionalizing nanocarrier, based on the 2D-ordered linear-chain carbon-based multicavity nanomatrix, containing the vacant functional nanocavities.

**Figure 5 nanomaterials-12-01041-f005:**
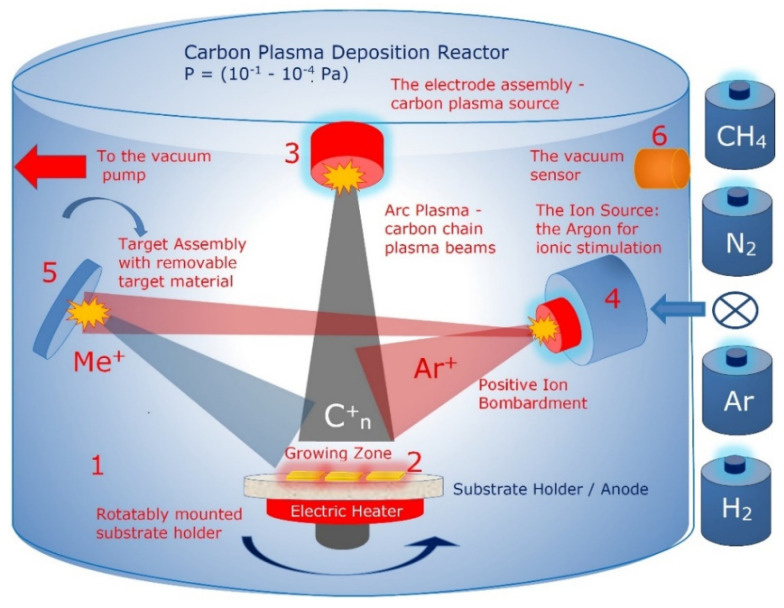
Schematic representation of the pulse-plasma deposition reactor for growing the 2D-ordered linear-chain carbon nanomatrix: 1—vacuum chamber; 2—substrate; 3—pulse-plasma carbon generator (graphite cylindrical main discharge cathode); 4—the ion source for ionic stimulation; 5—target assembly with removable target material; 6—vacuum sensor.

**Figure 6 nanomaterials-12-01041-f006:**
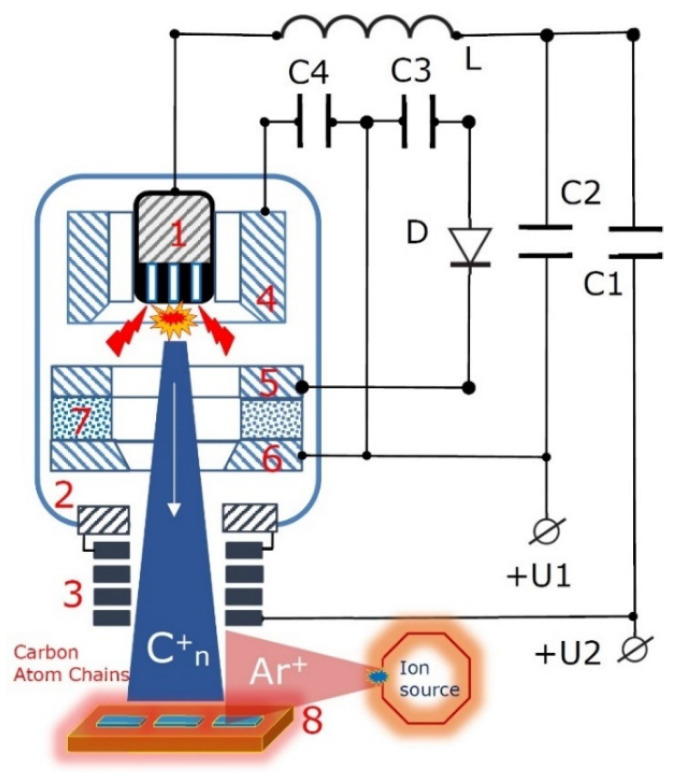
Schematic representation of the pulse-plasma generator installed in the reactor of the experimental set-up (Figure 5): 1—a cylindrical main discharge cathode (evaporated material, the high purity graphite) containing cylindrical rods manufactured from various materials and used for heteroatom doping; 2—the main discharge anode; 3—a solenoid final focusing system with plasma neutralization; 4—second auxiliary discharge anode; 5—ignition cathode; 6—ignition anode; 7—dielectric insulator; 8—substrate holder.

**Figure 7 nanomaterials-12-01041-f007:**
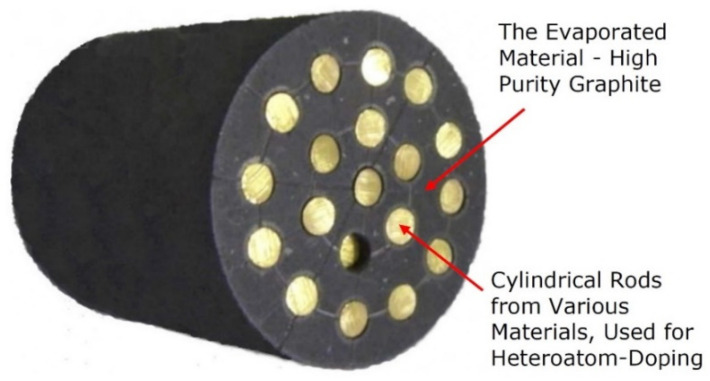
Example of design of a cylindrical main discharge cathode (evaporated material, the high purity graphite), containing cylindrical rods manufactured from various materials, used for heteroatom doping, for instance, silver, tungsten, gold, etc. This is item 1 in Figure 6.

**Figure 8 nanomaterials-12-01041-f008:**
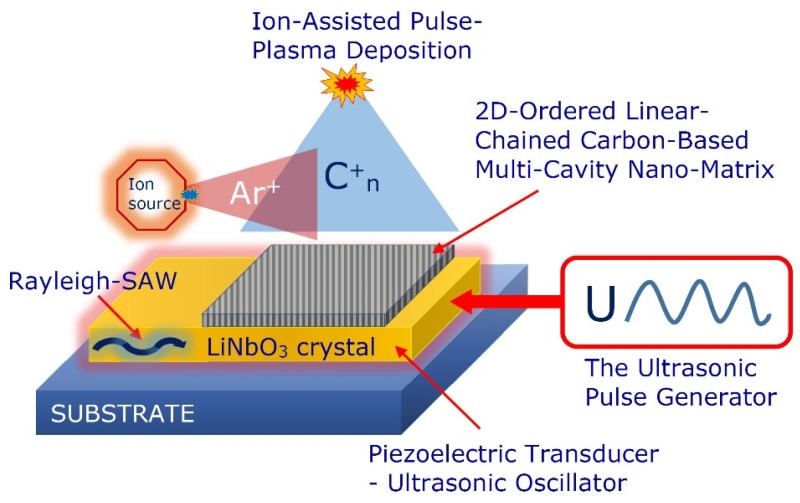
Schematic representation of using inverse piezoelectric effect during the nanomatrix ion-stimulated pulse-plasma deposition and schematic illustration of SAW streaming.

**Figure 9 nanomaterials-12-01041-f009:**
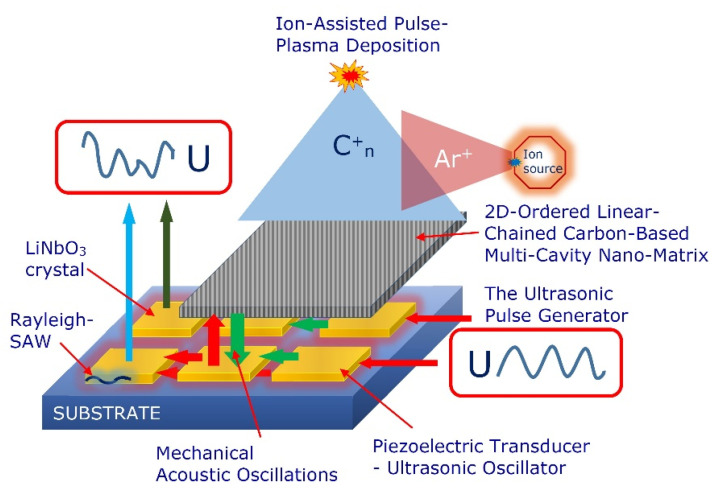
Schematic representation of using the direct and inverse piezoelectric effect at the nanomatrix ion-stimulated pulse-plasma deposition.

**Figure 10 nanomaterials-12-01041-f010:**
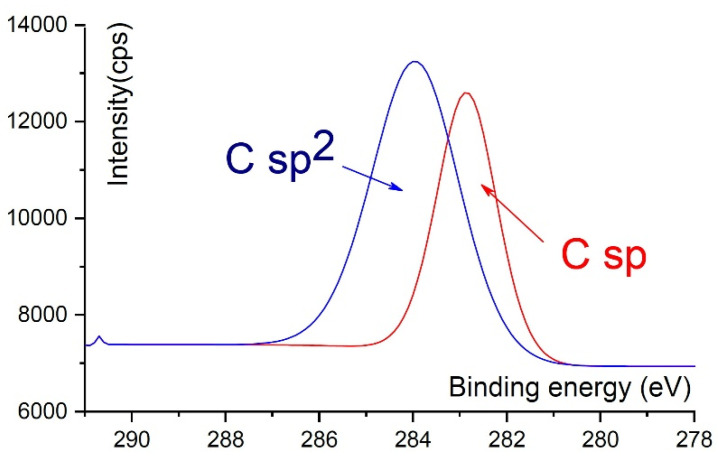
An example of the binding energy ratio for carbyne (sp) and graphene-like (sp2) hybridized bonds, obtained using X-ray photoelectron spectroscopy (XPS).

**Figure 11 nanomaterials-12-01041-f011:**
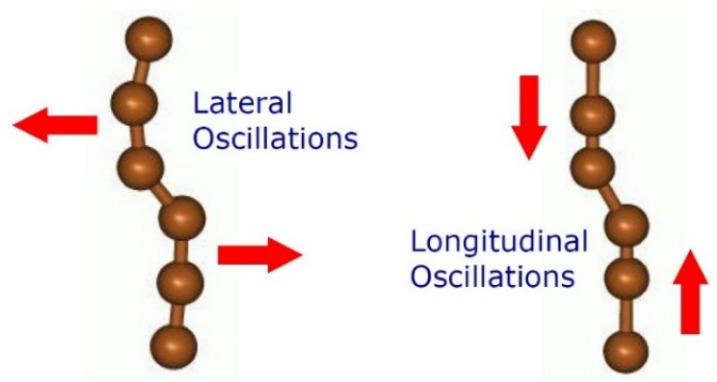
Schematic representation of the lateral and longitudinal oscillations of the sp-hybridized carbon chains.

**Figure 12 nanomaterials-12-01041-f012:**
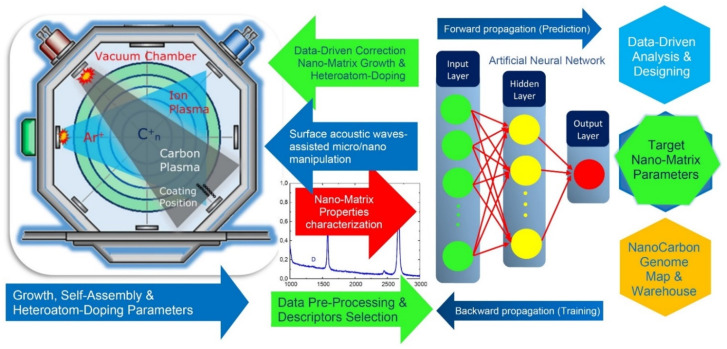
Machine learning-based fine-tuning of the functionalizing nanocarriers: a scheme of tracking and tailoring the key descriptors and linkages for incorporating into the data-driven carbon nanomaterials genome approach.

**Figure 13 nanomaterials-12-01041-f013:**
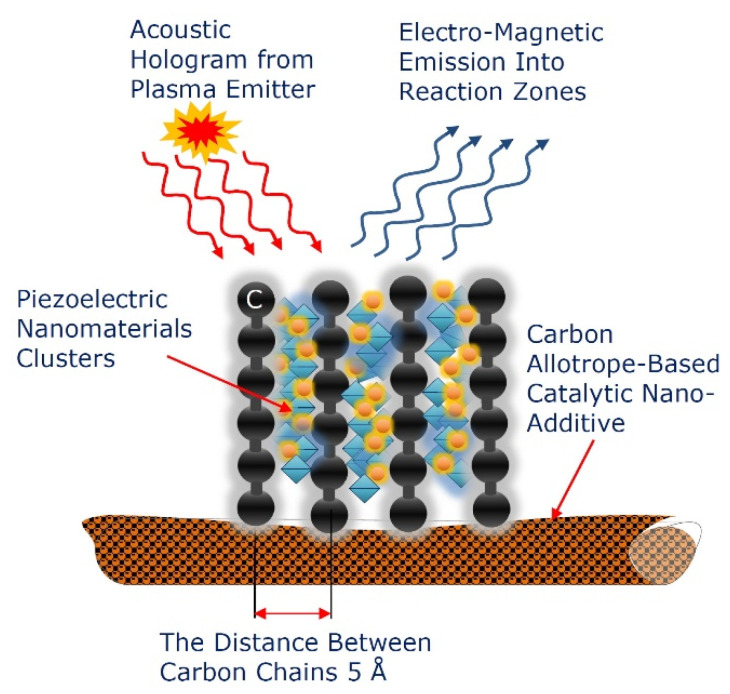
Schematic representation of the acoustic hologram transformation into the electromagnetic hologram through the 2D-ordered linear-chain carbon-based functionalizing nanocarrier, assembled by the piezoelectric nanomaterial clusters.

**Figure 14 nanomaterials-12-01041-f014:**
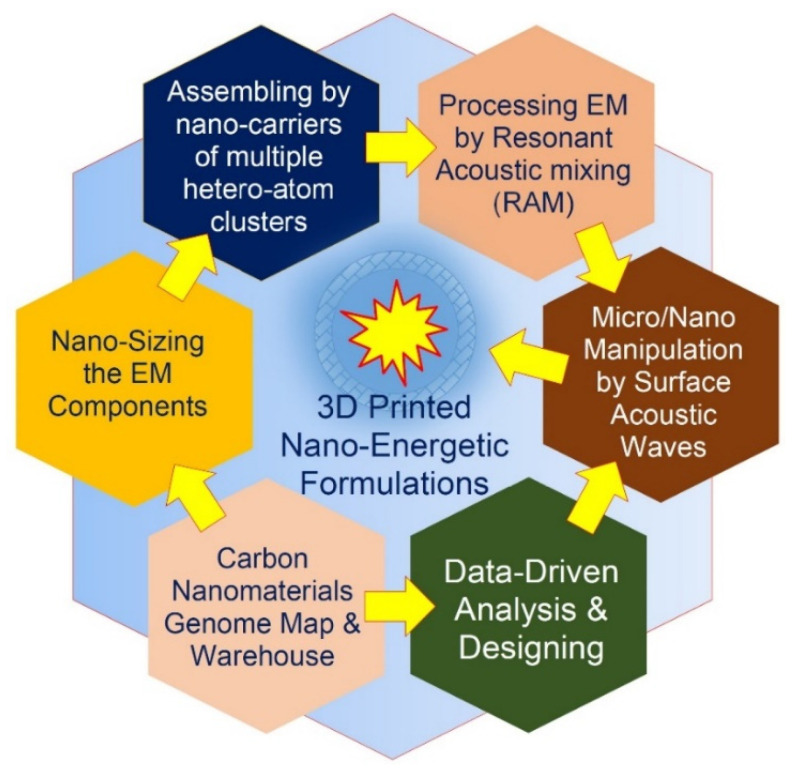
Modification of the technological chain of the transformative energetic materials’ multistage synthesis.

## Data Availability

Data available on request due to potential proprietary restrictions.

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
