# Peer review of "Tailoring Vibrational Signature and Functionality of 2D-Ordered Linear-Chain Carbon-Based Nanocarriers for Predictive Performance Enhancement of High-End Energetic Materials"

_nanomaterials, 2022, doi:10.3390/nano12071041_

Round 1

Reviewer 1 Report

The author reported that, “Tailoring the 2D-Ordered Linear-Chain Carbon-Based Nano- 2 Carriers Vibrational Signature and Functionality for Predictive Performance Enhancement of the Transformative Energetic Materials”,

“Tailoring the 2D-Ordered Linear-Chain Carbon-Based Nano- 2 Carriers Vibrational Signature and Functionality for Predictive 3 Performance Enhancement of the Transformative Energetic Materials”, The author should revise the title and make it more attractive.

The author should revise the abstract and make it small and attractive with a specific meaning.

The number of Keywords is meanless, so we suggested that the author should reduce keywords.

The introduction section should be revised, and the author added some specific views.

All figures make the same format.

The author should revise the Outlook and Perspectives and conclusions should be revised.

Reviewer 2 Report

I found the ideas presented in the manuscript very interesting and original. However it is a pity that the manuscript and the results are not presented in a good way. I found the structure of the manuscript not of a good quality and also the english style has not reached the required standard level for some parts of the text. There are some sections which can be considered fine but other are hard to understand. There some sentences of 4-5 lines, they are hard to understant and must be split. I propose to amend the manusript by considering the following comments and recommendations.

** Title: It is not clear and not simple. I propose to shorten it. If not you can replace it as follows: "Tailoring vibrational signature and functionality of 2D-ordered linear-chain Carbon-based nano-carriers for predictive performance enhancement of energetic materials".

** Manuscript structure: You have 11 sections. I propose to structure the paper in 4 ou 5 sections by fusing some sections and using subsections. You should precise the structure of the manuscript at the end of the introduction section (section 1), .i.e., indicate how the manuscript is organized, you gain in quality.

** Split long sentences by splitting them. It is better to use shorter sentences than long ones extending on 4 or 5 lines. Many parts should be rephrased. Try to improve the english style, the will let the reading more clear and easier.

** Uniformize or harmonize the used terms.

** Pay attention to verbs in connection with singular and plurial subjetcs.

** Some comments page per page.

*** P1

In lines 10-11. It is the concept which is considered as a pathway. Replace 'are' by 'is'. Remove 'the' from 'the game'

--L18 (and many other lines/pages). Replace nano-matrixes by nano-matrice

** P2

--L57. Insert ''have' after 'non-additives'

--L58. has-->have. Same for L70.

--L73. Recently has been proved-->Recently it has been proved that.

--L76. remarkable is not clear. Does it mean high, low ?

--L92-95. Long sentence. Please split it or rephrase it.

**P3

--L109. the next--> of next

--L127. Are you sure that the correct term is reducing? May be you mean reduced or reduction!

--L133. 'including hard to process materials' is not clear for me.

--L140. insert 'of' between 'generation' and 'electrically'

**P5.

--L153. matrixes-->matrices

**P5.

-- If I am correct, you did not mention Figure 1 in the text.

-- You use both carbyne and carbine. Is this right?

--L218. Insert 'a' before 'researc team'. have-->has.

--L220. Remove 'the' from 'the extremely'

--L234. inserting of--> insertion of

--L236-242. Please rephrase this part.

**P6

--L244. the room temperature--> room temperature

--L248. were found-->has found

--L252. for obtaining of the stable --> to obtain stable

--L253. based with growwing --> by growing

--L254. matrixes-->matrices

-- Figure 2, left panel. Why there is vertical chain of carbon atoms in the middle in addition to the six chains starting from the six vertices of the hexagon? Why do use different number of digits for the distance? (5.03 Angströms. Is the accuracy of the order of 1/100 of Angtrom? or 1/10?)

**P7.

--L282. Is carbyne-enriched correct or should be carbine-enriched?

--L287-292. Is it possible to show the vertical, longitudinal and transverse axes on a figure?

--L302. the new properties--> new properties

**P8

--L316. self-adjusts-->self-adjust

--L326. for control-->to control.

--L330. electric field applied-->applied electric field

--L337. at figure --> on figure or in figure

--L345. Use capital T for temperature sysmbol, small t is used for time.

**P9

--L355. presented at figure 6-->shown on figure 6

**P10

--L374. Remove 'The' from 'The example'.

--L388. the residual --> a residual

--L392. show the-->show that the

--L399-400. not reach more than 60-80-->does not exceed 60-80.

--L403. show the ion-assistance-->show that ion-assistance

--L404. sizes and spatial localization-->spatial dimensions and localization?

--L423-26. Please rephrase

--L431. the pulse-plasma--> of the pulse-plasma.

--L4335-36. You may say: 'Earlier, in papers [40-41] concerning atomic deposition experiments, the authors studeid ....". You may also use 'devoted to' instead of 'concerning'.

--L437.-38. You may say: 'Molecular dynamics simulations were used to describe the structuration...'

**P12

--L457. the paper -->paper

--L460. As were --> As was

--L486. Insert 'are' between 'vibration' and 'defined'

--L492. 'is usually including'--> usually includes

--L496. 'With taking into account that the 2D'--> 'Knowing that 2D' or 'Assuming that 2D'.

--L499. through using-->using

**P13

--L517. matrixes-->matrices

--L530. Line to remove

--L533. Remove 'At' from 'At applying'

--L538-41. To rephrase

**P14.

--L543. Remove 'the' from 'the 2D-ordered'

**P15

--L581. vibration-al- --> vibrational

**P16

--LL622. as effective--> as an effective

--L635. through-->by

--L637-42. To rephrase

--L652. matrixes--> matrices

--L654. The title may be written as follows: Data-driven tailoring of architecture and functionality of nano-carriers.

--L655. Remove 'the' from 'te research'

**P17

--L662. the data science--> of data science.

--L664. Insert 'to' between 'capacity' and 'effectively'

--L667. Application-->Applying

--L668. Insert 'of' after 'programming'

--L670-74. Please rephrase

--L680. the for-->for the

--L681. is correct selection the set'-->is the correct selection of the set'

--L685. 'use the' --> 'use of the'

--L692. Application-->Applying

--L699. 'presented....'-->can be found in a number of papers, e.g. [57, 59-63].

--L705. 'unstructured and unstructured'--> 'structured and unstructured'

**P18

--L718. at figure-->in figure

--L736. the-->of

--L737-39. To rephrase

--L742 'in external ...for manipulating'--> 'in a external ...to manipulate'

**P19

--L744-49. Long sentence. To rephrase

--L750. manipulating-->manipulation

--L751. Remove 'the' from 'the micro-'

--L771-80. To rephrase by splitting the long sentence.

--L781-85. To rephrase

**P20

--figure. Remove 'The' from 'The distance between Carbon...."

--L797. Insert 'an' between 'into' and 'the'

--L800. 'the plasma'-->'of plasma'. Remove 'te from 'the Teslaphoresis'

--L802. 'the new ways'-->'new ways'

--L803. 'opens the new'-->'opens new'

--L804. 'for the deep' --> 'of the deep'

--L812. 'the variety catalytic'-->'a variety of catalytic'

--L827. physics-chemical-->physicochemical

**P21

--L830. Insert 'of ' after 'Application'

--L842-43. for providing-->to provide

--L844, L848. functionalizing or functionalized? or both are correct?

--L853. approaches used-->used approaches

--L859-63. To rephrase

**P25. References [48] and [50] are identical.

Round 2

Reviewer 2 Report

The authors have amended their manuscript according to most of the comments. Even if the manuscript cas still be amended I think it is now acceptable for publication in this journal